# One CGIAR and the Integrated Agri-food Systems Initiative: From short-termism to transformation of the world's food systems

Bram Govaerts[1,2]☯*, Christine Negra[3]☯*, Tania Carolina Camacho Villa[1‡], Xiomara Chavez Suarez[1‡], Anabell Diaz Espinosa[1‡], Simon Fonteyne[1‡], Andrea Gardeazabal[1‡], Gabriela Gonzalez[1‡], Ravi Gopal Singh[1‡], Victor Kommerell[1‡], Wietske Kropff[4‡], Victor Lopez Saavedra[1‡], Georgina Mena Lopez[1‡], Sylvanus Odjo[1‡], Natalia Palacios Rojas[1‡], Julian Ramirez-Villegas[4‡], Jelle Van Loon[1‡], Daniela Vega[1‡], Nele Verhulst[1‡], Lennart Woltering[1‡], Molly Jahn[5‡]*, Martin Kropff[1‡]

1 International Maize and Wheat Improvement Center (CIMMYT), Texcoco, Mexico, 2 Cornell University, Ithaca, New York, United States of America, 3 Versant Vision LLC, New York, NY, United States of America, 4 Alliance CIAT-Bioversity, Cali, Colombia, 5 Jahn Research Group, University of Wisconsin-Madison, Madison, Wisconsin, United States of America

☯ These authors contributed equally to this work.
‡ These authors also contributed equally to this work.
* b.govaerts@CGIAR.ORG (BG); Christine@VersantVision.com (CN); molly.jahn@jahnresearchgroup.net (MJ)

**Data Availability Statement:** The manuscript does not contain datasets per se. Analysis of the

## Abstract

Agri-food systems are besieged by malnutrition, yield gaps, and climate vulnerability, but integrated, research-based responses in public policy, agricultural, value chains, and finance are constrained by short-termism and zero sum thinking. As they respond to current and emerging agri-food system challenges, decision makers need new tools that steer toward multi-sector, evidence-based collaboration. To support national agri-food system policy processes, the Integrated Agri-food System Initiative (IASI) methodology was developed and validated through case studies in Mexico and Colombia. This holistic, multi-sector methodology builds on diverse existing data resources and leverages situation analysis, modeled predictions, and scenarios to synchronize public and private action at the national level toward sustainable, equitable, and inclusive agri-food systems. Culminating in collectively agreed strategies and multi-partner tactical plans, the IASI methodology enabled a multi-level systems approach by mobilizing design thinking to foster mindset shifts and stakeholder consensus on sustainable and scalable innovations that respond to real-time dynamics in complex agri-food systems. To build capacity for these types of integrated, context-specific approaches, greater investment is needed in supportive international institutions that function as trusted in-region 'innovation brokers.' This paper calls for a structured global network to advance adaptation and evolution of essential tools like the IASI methodology in support of the One CGIAR mandate and in service of positive agri-food systems transformation.

different cases can be found here Maíz para Colombia https://repository.cimmyt.org/handle/10883/20218 Maize for Colombia: https://repository.cimmyt.org/handle/10883/20382 Maíz para México https://repository.cimmyt.org/bitstream/handle/10883/20219/60937.pdf?sequence=4&isAllowed=y and on the web.

**Funding:** This work was made possible thanks to the generous support of the Mexican Government through SADER in the frame of MasAgro, Crops for Mexico. The project is part of the CGIAR Research Program on Maize (MAIZE) and Wheat (WHEAT), which is generously supported by W1&W2 donors, including the Governments of Australia, Belgium, Canada, China, France, India, Japan, Korea, Mexico, Netherlands, New Zealand, Norway, Sweden, Switzerland, U.K., U.S., and the World Bank. Any opinions, findings, conclusions, or recommendations expressed in this publication are those of the authors and do not necessarily reflect the view of the donors mentioned previously. The funder provided support in the form of salaries for authors BG, CN, TCCV, XCS, ADE, SF, AG, GG, RGS, VK, VLS, GML, SO, NPR, JR-V, JVL, DV, NV, LW, but did not have any additional role in the study design, data collection and analysis, decision to publish, or preparation of the manuscript. The specific roles of these authors are articulated in the 'author contributions' section.

**Competing interests:** No competing interests. Versant Vision LLC has no competing commercial affiliation and therefore this does not alter our adherence to PLOS ONE policies on sharing data and materials."

# Introduction

## Agri-food systems in crisis

The world's agri-food systems are the sum of activities and relationships that determine how humanity's food is produced, processed, distributed, and consumed, together with the human, biological, chemical, and physical systems that shape these activities at every stage [1,2]. They are deeply interconnected with global trade and financial networks and operate across political borders.

Today, our agri-food systems are failing to deliver healthy diets to most of the world's population, even as food production methods–agriculture, foraging, and aquaculture–create widespread environmental harm from deforestation to resource over-exploitation to massive, planet-changing releases of activated nitrogen, phosphorus, greenhouse gases, and toxic chemicals. Globally, food production has increased with population growth, yet low-quality diets and food insufficiency leave billions of people suffering from malnutrition, undernutrition, and obesity [3]. Agricultural yield gaps (i.e., the difference between potential and actual crop and livestock productivity) remain pervasive [4,5]. The terrestrial and aquatic ecosystems that underpin food production have been heavily disrupted (e.g., carbon and nitrogen cycles; soil and water resources), contributing extensively to climate change [6–8]. Climate-related risks to agriculture (including synchronous crop failure across multiple regions) threaten all types of farmers as well as agri-businesses, financiers, insurers, consumers, and governments [9,10]. By any measure and at every scale, our agri-food systems are in urgent need of transformation to a condition where the food-related needs of every human are met completely and within planetary boundaries, consistent with a long term, livable future for all.

Simultaneously addressing the intimidating set of agri-food system challenges will require coordinated implementation of many strategies including rapid, extensive roll out of existing and emerging technologies (e.g., crop and animal breeding; resource use efficiency), demand management (i.e., aligning food consumption with health and environmental sustainability), ambitious spatial planning and target-setting (i.e., without double-counting or excess optimism), regulatory reform, and substantial financial investment [11,12]. Farmer adoption of sustainable agricultural practices and technologies and risk mitigation in change management will require context-appropriate combinations of agronomic support, value chain enhancement, tailored incentives, and responsive policies [13–16].

We have long understood that our complex and interconnected agri-food systems are fundamental to our collective aspirations for inclusive and equitable communities on a sustainable planet. Over the last seventy-five years, progress toward the post-war vision of a sustainable global agri-food system has been stymied when public and private sector decisions have been guided by short-term considerations and zero-sum thinking [17]. Economic policies, capital markets, commodity value chains, and even development programs rarely reflect an integrated systems approach [2].

The insufficiency of our current approach has been laid bare by global recessions and the COVID-19 pandemic. Across governments, civil society, and the private sector, stakeholders are embracing the global urgency to sustainably increase food production [18], in parallel with economic development and while adapting to increasing pressure from climate change and reversing natural resource degradation. Several of the UN Sustainable Development Goals (SDGs) address elements of sustainable agri-food systems [19]. However, the SDGs will not be achieved simply by promoting individual technologies, practices, or policies targeted to a single objective, nor can they be achieved primarily through digitization or financial inclusion [20].

Agri-food system solutions will require integrated systems approaches that go beyond introducing innovation through unilateral action and, to be inclusive, must embrace collaborative, research-based action by diverse stakeholders through a lens of system thinking [21–23]. To meet human needs under climate change and within resource limits, interventions will be needed across all components of food production and distribution value chains. Yet, many innovation and governance systems are weakly suited to handle current and emerging agri-food challenges [24,25]. Investment in supportive international institutions has not kept pace with the need for collaborative innovation focused simultaneously on productivity, health, and resilience [26–28]. The perennial impediment of weak local capacity for implementing context-specific solutions requires concerted effort to develop functional, inclusive civic and political institutions [29,30] and to improve scenario-based stakeholder planning [31].

To operationalize an integrated systems approach, agri-food system decision makers need new tools that help them to develop collaborative, research-based solutions, while accounting for socio-economic and political realities, without getting lost in complexity [32]. As an example of one such tool, this paper describes the Integrated Agri-food Systems Initiative (IASI) methodology. Based on insights gathered through development and validation of the IASI methodology in Mexico and Colombia, the authors call for the establishment of a structured global network to support use and adaptation of decision support tools embedded in systems thinking and in service of agri-food systems transformation.

## The Integrated Agri-food System Initiative

### Origins

The IASI methodology has conceptual origins in the global Knowledge Systems for Sustainability (KSS) research alliance that builds and tests knowledge systems to more sustainably manage complex risks related to food, energy, water, climate, and human security [33]. Methodologically, the IASI was inspired by the CSIRO-led Australian National Outlook (ANO). In 2015, the ANO presented an innovative analytical framework designed to allow stakeholders to explore plausible future scenarios, identify policy and investment strategies, and agree on near-term actions [34]. Focused on Australia's physical economy (i.e., water-energy-food nexus; materials- and energy-intensive industries), the 2015 ANO validated an integrated framework, which relied on loosely coupled biophysical, economic, and social models. In 2019, the ANO framework was applied to a broader set of stakeholder concerns (i.e., new technology- and science-based industries; energy and emissions; land use) [35].

Learning from the Australian experience, the International Maize and Wheat Improvement Center (CIMMYT), a CGIAR research center, developed and validated the Integrated Agri-food System Initiative (IASI), a holistic, multi-sector methodology that builds on diverse existing resources (e.g., datasets; models; approaches) to cultivate stakeholder agreement on coordinated public and private sector actions to enhance national agri-food systems. The IASI methodology is designed to engage agri-food system stakeholders from the public, private, and civil sectors such as representatives of farmers' associations, national research centers, universities, food and livestock feed companies, government agencies, and non-governmental organizations. Development of the IASI methodology was guided by the notion that agri-food system transformations depend on multi-scale, dynamic interplay among actors with different forms of power [36,37]. Understanding the variety of these interactions and processes, and their implications for governance, is critical for brokering broad stakeholder agreement on national policy and investment priorities [37], and for building upon on-the-ground experience with Agricultural Knowledge Management for Innovation [38].

## Methods

The starting point for applying the IASI methodology is to identify **windows of opportunity**, which will vary across national contexts. These might arise with a political transition, a fiscal crisis, a shift in trade conditions for key commodities, or a new donor initiative. In essence, these are times when influential actors are re-evaluating their goals and the means to reach them and are, therefore, more willing to entertain new approaches and information sources. Once a window of opportunity has been identified, the IASI methodology is applied sequentially. Major steps include:

1. Diverse experts examine the **current status** ("where are we today?") and the **business-as usual (BAU) scenario** ("where are we heading?") based on analysis of the socio-economic, political, and sectoral context and model-based projections.

2. Stakeholders determine a **preferred future scenario** ("where do we want to go?"), based on assessment of national implications, and define **drivers of change** toward a desired 2030 scenario, through a neutrally facilitated process.

3. Defined criteria are applied to stakeholder and expert inputs to **validate** drivers of change and to identify strategies and actions (e.g., public policies; value chain and market interventions; biotechnology applications) that can steer toward the preferred future scenario, which are then reviewed and **prioritized** by high-level decision makers.

4. Stakeholders agree on **measurable targets** and tangible, **time-bound actions** toward the preferred future scenario.

5. Stakeholders build shared commitment to a **tactical implementation plan** among traditional, non-traditional, and new partners.

6. Ongoing stakeholder engagement is organized around an **online dashboard** that tracks actions and progress toward targets and supports **course correction** and **coordinated investment**.

## Case studies

In 2017, the CIMMYT convened the Maíz para México (MpMx) initiative to support sustainable intensification of the Mexican maize sector, leading to the development of the IASI methodology. Subsequently, CIMMYT partnered with the International Center for Tropical Agriculture (CIAT), another CGIAR center, to adapt the IASI methodology to the Colombian context during a transitional socio-political moment, resulting in the Maíz para Colombia (MpCo) initiative. Table 1 summarizes the IASI processes in these two countries (further details can be found in the case studies in the Supporting Information and in public reports published by CIMMYT and CIAT [39,40]).

### Drive toward systems framing

In many countries, major commodity crops such as maize, wheat, or rice are natural gateways into national policy processes and constituencies because, as staple foods, they are often foundational to food security and national economies. While not a large economic driver, the enormous cultural and political importance of maize in Mexico, a staple food and a symbol of the country's heritage, has been amplified by the effects of free trade policies. In Colombia's post-conflict period, the under-developed maize sector was threatened by imports under open trade. In both Mexico and Colombia, the centrality of maize opened the door to robust

**Table 1. Development and adaptation of the Integrated Agri-food System Initiative through application to maize-based systems in Mexico and Colombia.**

| Maíz para México (MpMx) | Maíz para Colombia (MpCo) |
|---|---|
| A political or sectoral **window of opportunity** expands the scope of possible policy objectives. | |
| Late 2016 –A political transition and an uncertain trade context led Mexico's Agriculture Ministry to solicit CIMMYT's support with national agricultural planning. | Early 2018 –In a post-conflict period, CIMMYT and CIAT were invited to apply the IASI methodology to Colombia's under-developed maize sector, which was threatened by imports. |
| **1. Analysis of current status** (e.g., historical and contemporary data on the socio-economic, political, and sectoral context [41]) and a modeled **business-as-usual (BAU) scenario** are evaluated by an expert panel, which propose drivers of change for maize improvement by 2030, based on quantitative and qualitative inputs. | |
| Early 2017 –CIMMYT aggregated data, developed a 2030 BAU scenario, and convened an expert panel, which proposed five drivers of change. | Early 2018 –CIAT and CIMMYT aggregated data, developed a 2030 BAU scenario, and convened an expert panel, which proposed six drivers of change. |
| **2. Sectoral consultation** (with representatives from government, academia, and the agriculture sector), facilitated by a neutral, independent facilitator, determines a **preferred 2030 scenario** and preliminarily validates **drivers of change** and identifies corresponding strategies and actions. | |
| Early 2017 –CIMMYT convened an 85-person, multi-sectoral stakeholder workshop. (With CIMMYT's participation, an influential private sector organization undertook a parallel strategic planning process). | Mid-2018 –CIMMYT and CIAT convened a 60-person, multi-sectoral stakeholder workshop. (This event signaled renewed private sector participation in agricultural planning). |
| 3. Stakeholder and expert inputs are **validated** using defined criteria, resulting in a revised set of drivers of change, strategies, and actions to steer toward the preferred 2030 scenario, which are then reviewed and **prioritized** through structured one-on-one consultations with high-level representatives of diverse institutions. | |
| Mid-2017 –CIMMYT systematically reviewed expert and workshop inputs to produce two main strategies (each with underlying drivers of change, specified actions, and indicators), then undertook qualitative validation through one-on-one meetings with influential stakeholders. | Mid/late-2018 –CIMMYT systematically reviewed expert and workshop inputs to produce six revised strategies, then undertook qualitative validation through one-on-one meetings with diverse sectoral stakeholders. |
| **4. Measurable targets** and **tangible, time-bound actions** are proposed, based on the preferred 2030 scenario and prioritized strategies and actions, and anchored in SDG-related metrics (i.e., a global, multi-sector touchstone). | |
| Late-2017 –CIMMYT provided a detailed MpMx report to government leaders and sectoral actors. | Late-2018 –CIMMYT presented findings to executive and legislative political leaders. Mid-2019 –A CIMMYT/CIAT report presented MpCo 2030 targets and actions. |
| **5. Expanded partnership space** is created to translate shared commitments into a **tactical implementation plan** through purposeful stakeholder engagement. | |
| Mid-2018 –CIMMYT briefed Mexico's presidential candidates on MpMx. Early 2019 –The President's Office convened a high-level taskforce to propose MpMx implementation and policy integration. Early 2020 –The Secretary of Agriculture publicly launched MpMx [42]. | Early-2020 –National Agriculture Planning Office created an official plan based on the MpCo report. Early-2020 –CIMMYT and CIAT were mandated to convene a task force and explore potential funding collaborations. MpCo strategies and actions integrated into agricultural plans of farm groups and local governments. |
| **6. Online dashboards** are developed to monitor progress toward the preferred 2030 scenario (i.e., actions, targets), to support ongoing structured stakeholder engagement and **course correction**, and to display investible opportunities for donors, government, and the private sector. | |
| Mid 2019 –CIMMYT developed a project-level dashboard supporting ongoing monitoring. Mid-2019 –CIMMYT initiated a IASI process for wheat, beans, and cotton-based systems. Mid 2020 –Virtual stakeholder events in nine regions specified actions to be implemented. Mid 2020 –Donor-funded projects initiated. | Mid 2020 –CIMMYT initiated development of a project-level dashboard. End 2020 –Donor-funded projects initiated. |

engagement in the agriculture sector that, through the IASI methodology, enabled the emergence of a broader value proposition encompassing health and nutrition, national security, local economic development, and food self-sufficiency considerations. For example, as the

MpMx process rolled out, CIMMYT undertook a parallel process for native maize seeds and production practices (i.e., the autochthonous, diversified, maize-based *Milpa* production system) and a Beans for Mexico exercise was also initiated.

## Shared vision through design thinking and scenarios

The IASI methodology builds stakeholder consensus through design thinking (i.e., understanding specific needs to define an innovative solution), informed by situation analysis, modeled predictions, and scenarios for a discontinuous future. Current status is assessed through review of literature and historical trends (e.g., sub-national crop production; supply; consumption patterns; environmental conditions). A 2030 BAU agri-food scenario integrates analysis of biophysical changes in climate and crop production and socio-economic analysis (e.g., trade implications), in alignment with the timeframe of the UN Sustainable Development Goals. Loosely coupled models (based on spatially-explicit analyses and machine learning processes) are used to produce sufficiently accurate and precise scenarios that use metrics and narratives to make risks, benefits, tradeoffs, and counter-intuitive insights visible. While models and scenarios cannot predict the future, these tools can supplement the experience, knowledge, and intuition of agriculture sector stakeholders [43].

The BAU scenario is evaluated by an expert panel, composed of high-credibility specialists with diverse expertise (e.g., seeds; spatial analysis; climate change; trade) and track records of engaging outside their disciplinary realm. The expert panel provides deep technical knowledge throughout the IASI process. Multi-sector stakeholders are convened to review the BAU scenario and strategies proposed by the expert panel through a carefully designed, interactive workshop. (This workshop design was replicated by the Mexican Agriculture Ministry in development of a multi-crop strategic plan.) If the BAU scenario is perceived as positive by stakeholders, then no action is needed, but if an alternative future scenario is preferred, stakeholders identify strategies and actions that can steer away from the BAU scenario (which is commonly reinforced by near-term considerations or specialized interests) and orient toward preferred directions.

After further validation, consultations with mid- and high-level decision makers calibrate priority strategies and actions, while also building ownership and commitment among these influential leaders, who represent a diverse set of public and private institutions in the agriculture sector. While time-consuming, the goodwill developed through these consultations is instrumental for building novel collaborations promoting sustainability between government and industry (e.g., offtaker commitments) and continuity across political transitions. For example, in 2018, under a new presidential administration, MpMx was designated as a flagship project within the Crops for Mexico initiative. In Colombia, strong buy-in by an internal planning unit at the Ministry of Agriculture allowed the momentum of MpCo to continue as a new President and Agriculture Minister came into office in 2018.

## Multiple stakeholder entry points

The IASI methodology is designed to generate strategies, actions, and quantitative, SDG-aligned targets that have high likelihood of supportive public and private investment. It emphasizes timely provision of information and options (including estimated costs of inaction) to decision makers and enables multiple, coordinated entry points for stakeholders with different interests (e.g., policymakers; farm groups; financial institutions; input or service providers). In Mexico and Colombia, the IASI processes engaged traditional agricultural sector stakeholders as well as entities that are not usually pulled into technical agricultural

discussions, such as national development banks. To fill identified gaps, new entities such as alternative providers of rural finance were also engaged.

The IASI focus on drivers of change broadened the set of potential solutions and better embedded collectively identified strategies within government, increasing the likelihood of impact and continuity across different political regimes. In Mexico, the IASI process produced a tactical plan to improve agricultural production systems by translating innovation networks into visible knowledge co-creation infrastructure. These 'hubs' feature research platforms, demonstration modules, and extension areas where sustainable farming practices and technologies are tested, improved, and adapted with community participation [44,45]. This infrastructure provides a foundation for continuous scaling and enables timely regionalized impact assessments [38]. In Colombia, a hub network is being initiated, which will be critical to realizing the MpCo strategies. MpMx implementation will also be supported through the One CGIAR Excellence in Agronomy 2030 (EiA 2030) initiative, which will leverage data and analytics to deliver targeted digital agricultural advisory services (for farmers, farm advisors, and service providers) and to support government agencies and agricultural companies.

## Insights from IASI development and validation

### Beyond short-termism to an integrated systems approach

Building agreement around national policy change is a powerful scaling strategy for research-based solutions that contribute to sustainable agri-food systems. Too often, national policy processes are dominated by zero sum thinking and winner-take-all struggles (e.g., budget battles) that inhibit progress toward sustainability, equity, and profitability. Short-termism–excessive focus on short-term results at the expense of long-term interests–breeds internecine competition and politicization within commodity sectors, leaving little room for meaningful integration of new needs, such as climate adaptation, or evolving production practices to meet changing market demands [17].

Progress toward the SDGs requires facing up to the 'wicked' problems confronting agri-food systems [46]. As the seriousness of agri-food system crises is internalized by decision makers, a profound sense of urgency can lead to unsustainable or unilateral responses. For example, advanced technologies (e.g., sensors; crop protection; agricultural biologicals; artificial intelligence) are garnering investment based on anticipation of productivity and profitability gains that may not yet be substantiated by robust empirical evidence or without evaluation of potential tradeoffs, unintended consequences, or risks for equity and long-term sustainability [1]. Such responses to agri-food system crises may bypass more holistic solutions, grounded in multi-disciplinary science, that can generate cross-sectoral support and achieve multiple sustainability objectives (e.g., food security; poverty reduction; climate adaptation) [47–49].

By using existing tools to help sectoral stakeholders envision a different future and take action toward productive, inclusive, and resilient agri-food systems, the IASI methodology can steer away from technology over-reliance and winner-take-all approaches. The future-oriented and inclusive framing of the IASI methodology is essential to shifting stakeholder mindsets. By first collectively setting a vision and then linking it to present-day circumstances, constraints imposed by near-term concerns (e.g., budgets) and political dynamics can be minimized in favor of focusing on medium-term opportunities and cultivating trust that benefits can accrue to all collaborating stakeholders. The strategies and actions emerging from the IASI methodology reduce economic, reputational, operational, and policy risks faced by governments, global donors, and agricultural sector financiers by offering them a validated set of potential investments.

## Informing policy processes and coordinating action

While techniques for interactive, scenario-assisted, multi-objective development planning are not new [50–54], they have infrequently led to significant agri-food system change [31]. Challenges range from reductionist knowledge management systems [38], narrowly scoped outcome metrics [55,56] or technological feasibility analysis [47], poorly defined validation processes [43], and sparse attention to competing interests and policy incentives [14,32] to a lack of stakeholder engagement in scenario building [30] and tradeoff analysis [57,58] and weak governance and investment capacity [24].

Relative to other national agricultural planning approaches, the IASI methodology, as applied to maize-based systems in Mexico and Colombia, has demonstrated its effectiveness in overcoming typical barriers (e.g., short-term fixation on budgets; zero sum competition among stakeholders) and advancing multiple objectives simultaneously (e.g., improvement in crop yield and quality; farmer livelihoods; environmental protection). Importantly, both Mexico and Colombia underwent political transitions mid-way through the IASI process, yet there was substantial continuity in MpMx and MpCo.

One critical factor is the capacity to capitalize on critical moments such as the Mexican Agriculture Ministry announcing the first national agriculture planning process since 1966 (supported by the incoming president's interest to transform agriculture) and Colombia's recognition that its maize sector needed support in a post-conflict period. The IASI methodology does not replace national planning processes, rather it provides a framework for informing and focusing these processes. Facilitation that fosters consensus through candid, data-based discussions requires a neutral, independently positioned entity with skills and experience drawn from the arenas of business, research, political science and development.

As political and agriculture sector leaders (potentially in partnership with global donors) create or react to windows of opportunity, the neutrally positioned IASI methodology can strengthen stakeholder support for transformation of agri-food systems that expands productivity and long-term sustainability [45]. Emphasizing a collective process rather than a 'black box' solution, the IASI methodology is grounded in assessment of the current status and the BAU scenario and cultivates stakeholder agreement on how to pursue a more sustainable agri-food system. Leadership and accountability by influential stakeholders are key to creating the buy-in that unlocks important, but disaggregated data resources that are typically tightly held by public agencies, research institutions, and companies. A common vision for the future cultivates a cooperative mindset among stakeholders, enabling them to share rather than hoard datasets (and other sources of power asymmetry), allowing these data to become useful at a system level.

## Continuous improvement

Designed to promote thriving agriculture-based systems, the IASI methodology emphasizes mindset shifts toward sustainable and scalable innovation that responds to real dynamics of complex agri-food systems. It offers the possibility to simultaneously address multiple pressing development objectives, including unlocking the agricultural potential to adapt production systems to climate change, to sustainably manage land, soil, nutrient, and water resources, to improve food and nutrition security, and ultimately to reduce rural poverty by upscaling and mainstreaming results and actions.

Maturation of the IASI methodology will require continuous improvement. The new Crops for Mexico initiative expands on the IASI-mediated MpMx process and use of the IASI methodology is being explored in Africa and Asia. Future applications of the IASI methodology can pursue ongoing enhancements such as:

- Faster transition from crop-specific windows of opportunity to multiple food types (e.g., crops, livestock, and fisheries) and an integrated agri-food system framing (e.g., agricultural diversification; demand management and culture shifts toward healthy diets; building resilience through crisis response mechanisms).

- Increasingly sophisticated navigation of tradeoffs (e.g., higher yields vs. healthier diets) and constraints (e.g., energy; land; water) by simulating multi-dimensional outcomes for alternative policy directions.

- Greater reach (e.g., cross-sectoral; cross-institutional) and inclusivity (e.g., geographic balance; marginalized groups) and more diverse participation (e.g., consumer advocates; value chain actors; media).

- Deeper integration of system dynamics and climate change projections in scenario development and incorporation of flexible econometric and scenario planning models to generate a continuous range of options, rather than a limited number of scenarios.

- More explicit links to on-the-ground testing and implementation of strategies and actions (e.g., enhanced dashboards).

## International agricultural research centers as 'innovation brokers'

In developing the IASI methodology, CIMMYT leveraged knowledge gained through pre-existing international collaborations (e.g., CSIRO; KSS) and mobilized a diverse set of research-based approaches (e.g., situation analysis; loosely coupled models; scenarios). The opportunity to develop and test the IASI methodology arose because CIMMYT was an in-region institution that had built long-term trust relationships and demonstrated its capacity to adeptly mobilize data and knowledge toward technical and political challenges [59]. For example, recent collaboration on the MasAgro innovation system, which achieved yield increases among smallholder farmers and enhanced private sector value chains, had deepened CIMMYT's perception as a trusted partner of the Mexican government. In Colombia, CIAT had previously supported the government by producing national climate change projections.

Through the IASI processes in Mexico and Colombia, CIMMYT and CIAT functioned as 'innovation brokers,' enhancing their standing as trusted allies within national political frameworks. They created knowledge resources that informed subsequent politically-driven processes (e.g., in Mexico, MpMx findings and recommendations were integrated by the Agriculture Ministry and National Agriculture Council).

Deploying tools like the IASI methodology can amplify the ability of international agricultural research institutions to invigorate national policy development and demonstrate the business case for research-based solutions to public and private sector agri-food system decision makers [60]. With deep regional roots, these institutions are well-placed to influence the evolution of agri-food systems by guiding deployment of integrated strategies that combine breeding, genetics, agronomy, landscape management, enhanced nutrition, and other foundational elements of sustainable agri-food systems [26]. Through collaborative research initiatives, novel engagement mechanisms, and capacity building, the IASI methodology can catalyze improved availability of and access to technical support, information, technologies, and tools. For example, research undertaken by CIAT, CIMMYT, and national research centers is connected directly to farmers and adoption of improved technologies and practices by farmers simultaneously feeds back into research programs through on-farm data collection.

Historically focused on breeding for yield and calorie enhancement, international agricultural research institutions are increasingly producing multi-faceted solutions that enable

production landscapes to deliver human well-being and healthy diets within dynamic, global agri-food value chains [61–64]. With the advent of One CGIAR, enriched focus on agri-food system transformation will be supported by integration of capacities across regionally based centers and global programs [65].

## Recommendations

If our complex, interconnected agri-food systems are to meet human needs under climate change and within planetary resource limits, we must shift away from short-termism and zero sum thinking and toward integrated systems approaches. While new technologies, digitization, and other responses will be useful, they are unlikely to usher in inclusive, system-level transformation, which will depend on coordinated shifts in public policy, agriculture, value chains, and finance. As agri-food system decision makers respond to current and emerging agri-food system challenges, they will need new types of strategic planning tools that steer toward unconventional, evidence-based collaboration among diverse stakeholders.

### Use and adapt the IASI methodology

This paper presents a promising methodology for supporting integrated agri-food systems approaches that has been developed and validated within two national contexts. The IASI methodology leverages design thinking and scenario planning methods to generate a shared vision and broadly agreed solutions supported by agri-food system stakeholders. It is a data- and model-informed approach that facilitates collaborative identification of a preferred future by experts and stakeholders. The methodology fosters agility in determining implementation pathways through a tactical plan that translates proposed solutions into real integrated development programs and enables diverse multi-stakeholder contributions that align with public policy objectives.

In Mexico and Colombia, IASI processes are well underway (i.e., tactical plans are completed and resources are being mobilized) and further applications are anticipated in Africa and Asia, presenting opportunities for methodological refinement. To carry this work forward, several types of resources will be required: public and private financial investments; implementation capacity by dedicated program operation managers and consultants; and monitoring and evaluation specialists. As these applications mature, the IASI methodology will evolve to more rapidly and effectively engage diverse stakeholders in national agri-food system policy processes with enhanced technical capacity (e.g., a continuous range of policy scenarios) and implementation tracking.

### Invest in supportive international institutions

International agricultural research institutions can be indispensable 'innovation brokers' in regional agri-food systems. They are well-situated to lead application of the IASI methodology when windows of opportunity open. Through their long-term regional presence, high credibility, and mandate to translate research into sustainable development, these institutions can provide unique scientific leadership across political cycles, enhancing integration of sustainability considerations into national policy. Global donors have often found it expedient to commission international agricultural research institutions to produce outputs that are highly tailored to donor priorities [66]. If global donors are serious about helping developing countries to transform their agri-food systems, they will do more to unleash the essential leadership functions of trusted in-region research organizations [67]. Combined with the global inauguration of One CGIAR's enriched focus on agri-food system transformation through integrated regional programs, the IASI methodology will be a valuable tool.

### Mandate a global network for food systems transformation

Development and validation of the IASI methodology in Mexico and Colombia demonstrates the potential for a neutral, well-designed, science-informed, stakeholder-engaged process to create space for transformative innovation in national policy. Given the agri-food system crises made more visible during the COVID-19 crisis, tools like the IASI methodology are needed to support creation, design, and implementation of integrated strategies for healthy, resilient, equitable, and sustainable agri-food systems.

To propagate a robust body of knowledge and practice, a structured global network should be mandated by a high-level, multi-sectoral entity to systematize theory development, testing, validation, evaluation, and learning for agri-food systems transformation. By harmonizing disparate actions and accelerating continuous improvement of IASI and related methodologies, this network would engender a community of practice drawing from the KSS global alliance [68], One CGIAR communities of practice [69], EAT Forum [70], CSIRO [71], the Compact2025 Knowledge and Innovation hub [72], and other relevant groups. As One CGIAR regional programs are mobilized, a global food systems transformation network would serve as an integrating platform, enabling these programs to collaboratively develop and validate a shared set of best practices, to access technologies and services, and to co-design and co-implement projects with public and private sector partners.

A new global network for food systems transformation would support decision making in public policy, value chains, finance, and other components of agri-food systems. This network would harness diverse existing and emerging efforts toward a new field of research endeavor and global practice, analogous to the fields of business administration and organizational development, while accelerating methodological refinement and building capacity for further applications.

## Supporting information

**S1 File. Case study: Maíz para México–development and validation of the IASI methodology.**
(DOCX)

## Acknowledgments

We wish to thank all the participating stakeholders in the Crops for Mexico (Maize for Mexico) and Maize for Colombia Initiatives.

## Author Contributions

**Conceptualization:** Bram Govaerts, Christine Negra, Andrea Gardeazabal, Daniela Vega, Molly Jahn, Martin Kropff.

**Formal analysis:** Bram Govaerts.

**Funding acquisition:** Bram Govaerts.

**Investigation:** Bram Govaerts, Tania Carolina Camacho Villa, Xiomara Chavez Suarez, Anabell Diaz Espinosa, Simon Fonteyne, Andrea Gardeazabal, Gabriela Gonzalez, Ravi Gopal Singh, Wietske Kropff, Victor Lopez Saavedra, Georgina Mena Lopez, Sylvanus Odjo, Natalia Palacios Rojas, Julian Ramirez-Villegas, Jelle Van Loon, Daniela Vega, Nele Verhulst, Lennart Woltering.

**Methodology:** Bram Govaerts, Christine Negra.

**Project administration:** Bram Govaerts.

**Resources:** Bram Govaerts.

**Supervision:** Bram Govaerts.

**Validation:** Bram Govaerts.

**Writing – original draft:** Bram Govaerts, Christine Negra, Xiomara Chavez Suarez.

**Writing – review & editing:** Bram Govaerts, Christine Negra, Victor Kommerell, Nele Verhulst, Lennart Woltering, Molly Jahn.

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
