## [Decision Letter · Decision Letter 0]

26 Apr 2021

PONE-D-21-05705

One CGIAR and the Integrated Agri-food Systems Initiative: from short-termism to transformation of the world’s food systems

PLOS ONE

Dear Dr. Govaerts,

Thank you for submitting your manuscript to PLOS ONE. After careful consideration, we feel that it has merit but does not fully meet PLOS ONE’s publication criteria as it currently stands. Therefore, we invite you to submit a revised version of the manuscript that addresses the points raised during the review process.

The manuscript presents an interesting integrated approach with a case study and will get a lot of attention. However, it needs some revisions per the reviewer's comments. 

We look forward to receiving your revised manuscript.

Kind regards,

Abid Hussain

Academic Editor

PLOS ONE

Journal Requirements:

Reviewers' comments:

Reviewer's Responses to Questions

**Comments to the Author**

1. Is the manuscript technically sound, and do the data support the conclusions?

Reviewer #1: No

2. Has the statistical analysis been performed appropriately and rigorously? 

Reviewer #1: N/A

3. Have the authors made all data underlying the findings in their manuscript fully available?

Reviewer #1: No

4. Is the manuscript presented in an intelligible fashion and written in standard English?

Reviewer #1: Yes

5. Review Comments to the Author

Reviewer #1: General Comment:

This study covers an interesting topic on decision support tools for agri-food systems. Mainly, the authors introduce the Integrated Agri-food System Initiative (IASI) methodology as a such tool. This methodology has been applied to Mexico and Colombia as case studies. The authors described IASI methodology as a six-step process that needs to be applied together with stakeholders. The manuscript has a large focus on this methodology without a clear presentation of the results and the key messages of the study. Therefore, I suggest not considering the manuscript for publication. Additional reasons for this suggestion are follows.

Frist, the introduction section can further be strengthened by highlighting the gaps in existing tools. At the end of the introduction, the authors introduce IASI methodology is one of the decision support tools without introducing any other tools before.

Second, the manuscript does not present clearly the results of the applied IASI methodology. Further, looking the methodology it is not clear how decision makers can apply this methodology.

Third, the manuscript does not consist of discussion in the context of existing studies or tools, highlighting the novelties of this new methodology.

Fourth, the conclusions and recommendations of this study is not clear. It would be helpful for the readers when conclusions and recommendations are presented separately.

Fifth, please consider to use terminologies such as tools, methodology, and framework in a consistent manner. It would be better to use one term instead of multiple terminologies that say the same thing.

Specific Comment:

L83-86: The main message of this statement is not clear. Do the authors referring to the existing synergies and trade-offs among SDGs?

L141-1453: Who are these stakeholders? Why “2030” is mentioned here? It can also be 2050 or other years.

L152-153: Does this dashboard need to be online?

L184-186: Was that a systematic review?

L187-191: It is unclear what are these models. Are they crop models?

6. PLOS authors have the option to publish the peer review history of their article (what does this mean?). If published, this will include your full peer review and any attached files.

Reviewer #1: No

---

## [Editor Report · Decision Letter 1]

24 May 2021

One CGIAR and the Integrated Agri-food Systems Initiative: from short-termism to transformation of the world’s food systems

PONE-D-21-05705R1

Dear Dr. Govaerts,

We’re pleased to inform you that your manuscript has been judged scientifically suitable for publication and will be formally accepted for publication once it meets all outstanding technical requirements.

Kind regards,

Abid Hussain

Academic Editor

PLOS ONE

Additional Editor Comments (optional):

Thank you for addressing the reviewer's comments. Looking forward to the publication of this article.
---

## [Editor Report · Acceptance letter]

28 May 2021

PONE-D-21-05705R1 

One CGIAR and the Integrated Agri-food Systems Initiative: from short-termism to transformation of the world’s food systems 

Dear Dr. Govaerts:

I'm pleased to inform you that your manuscript has been deemed suitable for publication in PLOS ONE. Congratulations! Your manuscript is now with our production department. 

Kind regards, 

on behalf of

Dr. Abid Hussain 

Academic Editor

PLOS ONE